# Comparison of Behavioral Changes and Brain Activity between Adolescents with Internet Gaming Disorder and Student Pro-Gamers

**DOI:** 10.3390/ijerph17020441

**Published:** 2020-01-09

**Authors:** Ki Hyeon Kwak, Hyun Chan Hwang, Sun Mi Kim, Doug Hyun Han

**Affiliations:** Department of Psychiatry, Chung Ang University Hospital, 102 Heuk seok ro, Dongjak gu, Seoul 06973, Korea; letter2kh@naver.com (K.H.K.); mossflower28@naver.com (H.C.H.); cryslake@naver.com (S.M.K.)

**Keywords:** internet gaming disorder, pro-gamers, Child Behavior Check List, resting-state functional magnetic resonance imaging

## Abstract

While pro-gamers play according to defined living habits and planned schedules, adolescents with internet gaming disorder (IGD) exhibit irregular lifestyles and unregulated impulsive gaming behavior. Fourteen IGD adolescents and 12 pro-gaming students participated in this study. At baseline and after one year, demographic data, the Child Behavior Check List (CBCL), depressed mood, anxiety, and resting-state functional magnetic resonance imaging were assessed. Over the year, IGD adolescents played games as per their usual schedule, while pro-gamer students played according to their school’s team schedule. After one year, the pro-gamers’ scores had decreased in the CBCL-total (total problematic behaviors), CBCL-externalizing (under-controlled behavior, like impulsivity and aggression), and CBCL-internalizing (over-controlled behavior like depression and anxiety) compared to those of the IGD adolescents. Both groups displayed increased brain activity in the parietal lobe (a component of the attention network) over the years. Compared to pro-gamers, IGD adolescents showed higher brain activity within the left orbitofrontal cortex. Brain activity within the orbitofrontal cortex was associated with CBCL-externalizing scores. These results suggest that gaming had increased the attention network’s brain activity, but a well-organized support system could lead to different results, in terms of improved behaviors and suppressing brain activity within the orbitofrontal cortex.

## 1. Introduction

### 1.1. Debates on Internet Gaming Disorder

Over the last two decades, many studies have suggested that excessive and problematic online gaming should be defined as a psychiatric disorder along with other addictive behaviors and functional impairments [1,2,3]. In a national survey of 1178 Americans aged between 8 and 18 years old, Gentile et al. [4] reported that patients with internet gaming disorder (IGD) performed poorly at school and suffered from attentional problems. Stockdale and Coyne [5] asserted that patients with IGD had poor mental health and cognitive functions, including poorer impulse control and attentional problems, compared to healthy controls. Hull et al. [6] reported that mature-rated, risk-glorifying gameplay was associated with substance use (including alcohol and cigarettes), aggression, delinquency, and risky sex. Since 2014, the World Health Organization (WHO) has regarded internet gaming addiction as a significant public health problem. In 2018, the International Classification of Diseases version 11 (ICD-11) defined gaming disorder as a medical disease with a pattern of repetitive or persistent gaming behavior [2,3].

Several studies have suggested that the psychobiological mechanisms underlying behavioral addiction resemble those of chemical addiction, such as that of alcohol and other drugs [7]. Moreover, it is believed that those with IGD share similar characteristics in neural activity and cognitive dysfunctions with those with a gambling disorder [8].

Based on previous correlational studies, the brain neurobiology of IGD can be summarized as increased activity within the orbitofrontal cortex and decreased activity within the dorsolateral prefrontal cortex [9,10]. Kim and Kang [9] found that IGD patients, compared to other gamers, showed a stronger functional connectivity (FC) within the orbitofrontal cortex, which is involved in motivational salience, and a decreased FC within the dorsolateral prefrontal cortex, which is involved in learning and attention. Additionally, Wang et al. [10] reported a decreased cortical thickness within the dorsolateral prefrontal cortex, related to cognitive control, decision making, and reward and loss processing. Motivational salience was thought to be associated with continuous and repetitive behaviors in various addictive diseases [11]. The dorsolateral prefrontal cortex was reported to control the orbitofrontal cortex [12]. In many addictive behaviors, the dorsolateral prefrontal cortex can fail to control the hyper-activated orbitofrontal cortex [12].

However, there is some debate about whether IGD or gaming disorder can be classified as a medical disease [1,13,14]. The American Psychiatric Association (APA) included IGD in Section III of the Diagnostic and Statistical Manual of mental disorders (DSM-5), as it requires further research and data accumulation, due to a lack of IGD cases, high prevalence of non-formal diagnostic criteria, and an unbalanced prevalence rate [1]. Several studies have also suggested that IGD might just be a social phenomenon induced by environmental and social stress [13,14]. Jeong et al. [13] suggested that adolescents excessively played internet games when they were under academic stress or lost self-control. The period of adolescence was known as a particularly at-risk developmental stage for problematic internet game playing [15,16], because individuals were prone to impulsivity and uncontrolled, unplanned internet game play [16,17]. Large, accumulating literature suggests that the prevalence of IGD is dominant in the male population [16]. The initial onset of IGD is not associated with gaming time or the game’s genre [13]. Pang et al. [14] reported that gaming motivation in adolescents with IGD was associated with social anxiety and psychological stress.

Several studies expressed opposing opinions regarding the cognitive dysfunction induced by internet gameplay [18,19,20]. Latham et al. [18] asserted that video gaming could result in extensive improvements in various cognitive functions. Pallavicini et al. [20] suggested that video games could be used as a tool to improve the patient’s wellbeing through cognitive and emotional training. Mentiplay et al. [19] also showed that video games could help improve symptoms of patients with developmental coordination disorders. Ballesteros et al. [21] suggested that the working memory and selective attention of older adults could be improved through video games.

### 1.2. Pro-Gamers vs. Patients with IGD

It’s already known that IGD patients not only display excessive internet gameplay, but also have unplanned and irregular lifestyles, and impulsive, unregulated behavior in the absence of support systems [5,22,23]. Kim et al. [24] reported that most IGD patients were unable to control their impulses and displayed delinquent behavior. In a two-year follow up study, Baysak et al. [22] reported that social support systems could be a protective factor for IGD.

There is another population in South Korea that also displays excessive internet gameplay but does not meet the criteria of IGD, called professional gamers (pro-gamers). Pro-gamers belong to the internet game league and have contracts with teams that result in a salary and potential prizes. To perform well, pro-gamers practice for about 10 h a day within a defined schedule [25]. This schedule includes practicing, physical exercise, team strategy conferences, resting, and mealtimes.

For several years, A-hyun High School has recruited students who want to become professional gamers. Before going to A-hyun High School, all adolescents must complete the second grade at a general high school. In their schooling, they have a similar curriculum to third grade general high school students, such as Korean, English, mathematics, and sciences. In addition, they also play internet games for 4–5 h a day, following a schedule the game teacher suggests.

### 1.3. Resting-State Functional Magnetic Resonance Imaging (MRI) and Fractional Amplitude of Low-Frequency Fluctuation

Resting-state functional MRI (rs-fMRI) is thought to measure spontaneous brain activity, representing brain function [26]. Low-frequency (0.009–0.08 Hz) fluctuations (ALFF) of the blood oxygen level-dependent signal were thought to be related to spontaneous neural activity in rs-fMRI [27]. The fractional amplitude of low-frequency fluctuations (fALFF) is an advanced version of the original ALFF and helps to detect spontaneous brain activity more sensitively [28]. Changes in the fALFF have already been reported in several studies of psychiatric diseases, including schizophrenia [29], autism [30], and attention deficit hyperactivity disorders (ADHD) [31]. Moreover, Kim et al. [32] observed correlations between changes of fALFF within the inferior frontal gyrus and changes in delinquency and externalizing behaviors.

### 1.4. Hypothesis

We hypothesized that long term internet game play would increase the brain activity within the attentional system in both groups. However, the existence of a support system, including a regular schedule and a supervisor, would lead to different results in terms of behaviors and brain activity. Student pro-gamers with a good support system would show improved behavioral scores compared to IGD patients. In addition, a good support system would prevent the hyperactivity within the orbitofrontal cortex in student pro-gamers, while a poor support system would not prevent hyperactivity within the orbitofrontal cortex in response to impulsive internet game play.

## 2. Methods

### 2.1. Participants

The participants in the current study were classified into two groups; pro-gamer students and IGD adolescents. The two groups were both engaged in excessive internet game play. However, the pro-gaming students with a support system had characteristics of a regular lifestyle, while IGD adolescents, without a support system, showed an irregular lifestyle.

The institutional review board of the Chung-Ang University Hospital approved the research protocol for this study. All adolescents were informed about the study’s procedures and signed a written informed consent form. Their parents also provided written informed consent. The diagnostic criteria of IGD were based on the DSM-5 [1].

From September 2016 to December 2017, 121 IGD adolescents visited the Department of Psychiatry at the OO University hospital for diagnosis and treatment. In contrast to the systematic caring group (student pro-gamer group), IGD patients who visited for an initial assessment but received no treatment (non-systematic caring group) were regarded as the compared group.

Of the 121 IGD adolescents, we found 27 adolescents who completed psychological and brain-imaging assessments at their first and second visits, but they had not received any treatment or interventions, such as cognitive behavior therapy or psychiatric medications, over the years. Although parents and caretakers had asked that the adolescents receive treatment for IGD, those adolescents refused treatment due to no interest in treatment, no insight of problematic behaviors, and laziness to come to the treatment center. On January 2018, we contacted 27 adolescents via phone to introduce our study, and 14 adolescents and their parents agreed to participate. The other 13 adolescents or their parents were not willing or able to participate in study.

In 2017, 55 adolescents who wanted to be professional gamers applied to A-hyun High School’s Pro-Gamer Department. With a ranking from the internet game “League of Legend” competition and a basic academic ability test, 12 adolescents qualified and were accepted to be admitted to the school. After listening to the purpose of our research, all pro-gaming students and their parents agreed to participate in our study.

The pro-gamer students had school schedules, including regular academic classes (4 h/day), physical sports class, strategy meetings, mealtimes, and game training time (3 h/day in school). Two teachers managed and checked the schedules, while the parents of the IGD adolescents observed the life patterns of the adolescents and reported it. At a baseline and after a year, both groups were asked to give their demographic data, including age, educational year, and internet gameplay time (“How many hours per day do you play internet game?”) as well as a number of psychological scales, including the Young Internet Addiction Scale (YIAS), Child Behavior Checklist (CBCL), Child Depressive Inventory (CDI), Beck Anxiety Inventory (BAI), and Korean ADHD Rating Scale (K-ARS). Resting-state functional MRI (rs-fMRI) was also undertaken.

### 2.2. Clinical Scales

The CBCL is known as a screening tool for assessing problem behaviors in children and adolescents based on the parent’s self-report [33,34]. The Korean version of the CBCL (K-CBCL) has standardized reliability and validity [33,34]. Parents assessed their children and adolescents, aged between 4 and 18, using the K-CBCL, in terms of social adaptation and problem behavior. It consisted of 117 questions, with three subscales including a total problem score, as well as externalizing and internalizing scores. Higher scores indicated a greater degree of behavioral and emotional problems [33,34]

The Young Internet Addiction Scale (YIAS), proposed by Young in 1998, is a self-reporting measure for routine internet use [35]. The YIAS consists of 20 self-assessment questions, each graded on a scale of 1 to 5 (“rarely” to “always”). YIAS scores above 50 are considered to reflect problematic internet use. The Korean version of YIAS was verified by Lee et al. The YIAS’ internal consistency has been reported to be in the range of 0.90 to 0.91 [36].

The Children’s Depression Inventory (CDI), developed in 1977 by Kovacs (1985), is a self-reported measure of depression in children and adolescents aged 7 to 17 years old [37]. The 27 items of the Korean version of CDI, with internal consistency of Cronbach’s α = 0.88, was verified by Cho and Lee [38].

The Beck Anxiety Inventory (BAI), with 21 questions, is used to measure anxiety severity [39]. The BAI is scored on a scale of 0 to 3 and has a maximum score of 63 points. The Korean version of the BAI, with an internal consistency of Cronbach’s α = 0.93, was verified by Kwon et al. [40].

The Korean ADHD Rating Scale (K-ARS) is an ADHD symptom severity scale composed of 18 items (9 items for inattentive evaluation and 9 items for hyperactivity evaluation) designed by Dupaul [41]. The Korean version of the ARS, with an internal consistency of 0.77 to 0.89, has been verified by So et al. [42].

### 2.3. Brain Image Acquisition and Processing

All MRIs were acquired using a 3.0 T Philips Achieva scanner. All participants laid down with their eyes closed and were asked to stay awake. The heads of the participants were stabilized with cushions and taped for severe head movement prevention. Resting-state (Rs-fMRI) images were acquired axially, with an echo-planar imaging sequence, using the following parameters: TR/TE = 3000/40 ms, 40 slices, 64 × 64 matrix, 90° flip angle, 230 mm field of view (FOV), and 3 mm section thickness without a gap. Each scan lasted 720 s, and 230 volumes were obtained. The first 10 volumes were removed for gradient field stabilization.

Data preprocessing and processing were carried out using the Data Processing Assistant for Rs-fMRI (DPARSFA-http://www.restfmri.net), which is a plug-in software that works with Statistical Parametric Mapping (SPM12; http://www.fil.ion.ucl.ac.uk/spm/software/spm12/) and the Rs-fMRI Data Analysis Toolkit (REST; http://resting-fmri.sourceforge.net). Images were corrected for slice acquisition time differences, realigned, normalized, spatially smoothed with a 6 mm full-width half maximum kernel, de-trended, and temporally band-pass filtered to 0.01–0.08 Hz. Based on the results from the realignment processing by SPM, subjects that had a translation or a rotating motion greater than 3 mm or 2°, respectively, in any direction, were excluded from the study. No subject was excluded because of excessive head motion.

To assess brain activity among the groups at baseline, fALFF was performed using the REST software before treatment was administered. During preprocessing, Fisher-transformed correlation coefficients were measured for each pair of the regions of interest (ROIs) in each participant. The fALFF between the ROIs was calculated using the CONN-fMRI FC toolbox (version 15; https://www.nitrc.org/projects/conn). The fALFF method was used to find the regions where the local connectivity was correlated to the clinical scores. As an indicator of fALFF value, Kendall’s coefficient of concordance of a given voxel was calculated using the surrounding 26 voxels to evaluate the similarity of the time series. These were then standardized using Z-scores to perform group analyses.

### 2.4. Statistics

Demographic and psychological data between the pro-gamer group and the IGD adolescent group were analyzed using the Mann–Whitney U test. The changes in the psychological scales were assessed with the Kruskal–Wallis test. The differences in the psychological scales’ changes were also evaluated with an analysis of variance (ANOVA).

At baseline, the fALFF between the student pro-gamers and the IGD adolescents was compared using an independent *t*-test with the SPM12 software package. We performed a paired *t*-test, using the SPM12 software, to investigate fALFF changes in both the student pro-gamers and the IGD adolescents. Additionally, the difference in fALFF changes between the student pro-gamers and IGD adolescents were measured with an ANOVA using the SPM12 software package. The correlation was calculated between the fALFF map and the CBCL using SPM12. The resulting maps were set to a threshold using a *p*-value of <0.05, and false discovery rate correction was made for multiple comparisons with an extent of more than 20 contiguous voxels.

## 3. Results

### 3.1. Comparison of Demographic and Psychological Data

There were no significant differences in age, intelligence quotient (IQ), internet gaming time, CDI, BAI, K-ARS, CBCL-T, and CBCL-E scores between the student pro-gamer and IGD adolescent groups. However, the IGD adolescents showed increased YIAS and CBCL-I scores compared to the student pro-gamer group (Table 1).

After a year, no difference was seen in the YIAS (F = 1.12, *p* = 0.30), internet game playing time (F = 0.62, *p* = 0.44), CDI (F = 3.50, *p* = 0.07), BAI (F = 0.02, *p* = 0.89), and K-ARS (F = 0.46, *p* = 0.51) scores between the two groups. However, the CBCL-total scores (F = 12.76, *p* < 0.01) and CBCL-externalizing (F = 19.81, *p* < 0.01) and CBCL-internalizing (F = 11.09, *p* < 0.01) scores decreased in the student pro-gamer group but did not change in the IGD adolescent group (Figure 1).

The Children Behavior Check List (CBCL) total (F = 12.76, *p* < 0.01), CBCL-externalizing (F = 19.81, *p* < 0.01), and CBCL-internalizing (F = 11.09, *p* < 0.01) scores in student pro-gamers decreased while all CBCL scores in IGD adolescents were unchanged.

### 3.2. Comparison of the Changes in fALFF between Student Pro-Gamers and IGD Adolescents after a Year

At baseline, there were no regions with different brain activity at resting state between the IGD adolescents and the student pro-gamers.

After a year, both groups showed increased brain activity within the attention networks (parietal lobe). The details were as follows: The fALFF within the parietal lobe (x, y, z, 42, −66, 36, voxels = 26, T = 4.61, uncorrected *p* < 0.001) in the student pro-gamer group and the fALFF within the parietal lobe gyrus (x, y, z, 36, −24, 45, voxels = 25, T = 4.52, uncorrected *p* < 0.001) in the IGD adolescents increased.

Only in the IGD adolescents, orbitofrontal cortex activity increased after a year. The details are as follows: IGD adolescents showed an increased fALFF within the left orbitofrontal cortex, including the left subcallosal gyrus (x, y, z, −6, 12, −12, voxels = 121, T = 6.37, uncorrected *p* < 0.001), left orbital gyrus (x, y, z, −15, 33, −24, voxels = 121, T = 5.99, uncorrected *p* < 0.001), and left inferior frontal gyrus (x, y, z, −21, 27, −21, voxels = 121, T = 6.37, uncorrected *p* < 0.001), compared with those of the student pro-gamers (Figure 2).

### 3.3. Correlation between the fALFF and CBCL Scores in All Adolescents (Student Pro-Gamers and IGD Adolescents)

The fALFF values within the left inferior frontal gyrus were associated with the CBCL-externalizing scores in all adolescents (r = 0.50, *p* < 0.01). However, there were no significant correlations between the fALFF values within other areas and the CBCL-total or CBCL-internalizing scores.

## 4. Discussion

This study compared two groups under the same conditions of excessive internet gaming, one with a support system in place and one with no support system. Even though both groups played internet games for more than seven hours a day, it was assumed that the existence of a support system to keep a regular schedule would lead to different results in terms of behaviors and brain activity. The student pro-gamer group showed improved behavioral scores after a year compared to the baseline. However, IGD adolescents showed no improvement in behavioral scores and the impulse control network showed dysfunctional brain activity.

### 4.1. Mproved Problematic Behavioral Scores in Student Pro-Gamers Compared to IGD Adolescents

At baseline, CBCL-internalizing scores in the IGD adolescents were higher than those observed in student pro-gamer group. In addition, the student pro-gamer group showed improved behavioral scores as assessed using the CBCL, compared to IGD adolescent. Moreover, the behavioral score improvements on the CBCL in student pro-gamers included both the CBCL-externalizing and internalizing scores. The CBCL was designed to measure the degree of behavioral and emotional problems in children [43]. The CBCL-externalizing score shows the degree of external problems, including social, thought, and attentional problems, while the CBCL-internalizing score shows the degree of internal problems, including anxiety, depression, and somatic complaints [43]. Additionally, the CBCL-total scores have been suggested as primary screening instruments for ADHD in Korean children [44]. The CBCL-internalizing scores were positively correlated with the Beck Depressive Inventory scores in patients with mood disorders [45]. Altogether, our CBCL results may suggest that the student pro-gamer group showed an improvement in their behavioral and emotional status. With the CBCL results at baseline and follow up, a different interpretation can be suggested. The pro-gamer group was preparing for a pro-gamer career, while the IGD adolescent group had no clear future direction. This situation may have led to the bias of the parental assessment in the CBCL.

Many studies have reported the effects of gaming on adolescent behaviors [46,47]. Greitmeyer et al. [46] suggested that video gaming would affect the social behaviors of gamers, especially when violent and pro-social video games are considered. Shao et al. [47] asserted a positive correlation between the use of violent video games and adolescent aggressiveness. However, our results may indicate that environmental factors and the existence of a support system affects the gamers’ behaviors to a greater extent than the game play itself. In a large sample study of adolescents who play violent video games, Przybylski [48] reported that there was no significant correlation between violent video games and aggressive behavior in adolescents. In a survey conducted by the US National Research Council, environmental factors, including family resources and school quality, were seen as crucial factors in preventing mental, emotional, and behavioral disorders in young people [49].

### 4.2. Increased Brain Activity within the Attention Network (parietal lobe) in Response to One-Year Internet Gameplay in Both Groups

After playing internet games over a year, brain activities within the parietal lobe in both groups increased. The parietal lobe is known as a part of the attention network in the human brain [50]. In previous IGD studies, gaming was seen to affect brain activity within the attention network [25,51,52]. Action video gamers made faster and more precise responses toward targets, using enhanced attention skills [52]. Compared to non-gamers, the frontoparietal network in gamers was used to a greater extent in responding to attention-demanding tasks [51]. Interestingly, repetitive and impulsive internet gaming was thought to be a self-medication for children with ADHD [53,54]. Evren et al. [53] have reported that ADHD was associated with the severity of internet addiction and IGD among university students. Eight weeks of methylphenidate treatment in ADHD children with internet addiction improves ADHD symptoms as well as the severity of internet addiction [55]. Conclusively, increased brain activity within the parietal lobe in both student pro-gamers and the IGD group was associated with internet gaming.

### 4.3. Increased Brain Activity within the Orbitofrontal Cortex of the IGD Adolescents in Response to a Year of Internet Gaming

Increased brain activity, represented by the fALFF value, within the orbitofrontal cortex was only observed in IGD adolescents in this study. The imbalance between the dorsolateral prefrontal cortex and the orbitofrontal cortex, due to a hyper-activated orbitofrontal cortex, was thought to be related to cognitive control and decision-making dysfunctions, as well as impulsive behaviors in patients with gambling disorders [55]. In the seed base FC analysis, Kim et al. [24] reported the imbalance of FC from the orbitofrontal cortex to the dorsolateral prefrontal cortex in IGD patients. In internet addiction, functional dis-connectivity between the orbitofrontal cortex, dorsolateral prefrontal cortex, and anterior cingulate, due to hyper-activated orbitofrontal cortex, are associated with impulsive and delinquent behaviors [56].

### 4.4. Association between fALFF Values within the Orbitofrontal and the CBCL-Externalizing Scores

The association between impulsivity, drug seeking behaviors, and dysregulation of the orbitofrontal cortex has been continuously reported in substance use disorder [57,58]. The hyper-activated orbitofrontal cortex might aid to balance the FCs between the dorsolateral prefrontal cortex and the orbitofrontal cortex [11,12].

CBCL-externalizing scores were thought to be associated with impulsive behaviors in children and adolescents [44]. Eisenberg et al. [59] described that problem behaviors in children were related to negative feelings or impulsivity. In other words, difficulty in controlling one’s behavior is related to impulsiveness [60].

As the student pro-gamers had a support system in place, this may have helped to promote planning and regular lifestyles. Planning and impulsivity in human behavior have been seen to be closely associated with the functioning of the dorsolateral prefrontal cortex [61,62]. Conclusively, a well-controlled support system that includes planning and a regular life pattern may improve the balance of brain activity between the dorsolateral prefrontal cortex and the orbitofrontal cortex in student pro-gamers.

### 4.5. Limitations

There were several limitations to our study. First, the number of participants was too small to generalize our results. Because of the small numbers, the brain analysis results have been presented in an uncorrected format. Second, there were few assessment tools used to assess potential environmental factors in our study. Future studies should include a larger number of participants and evaluate the environmental conditions. Third, the assessment tool for excessive internet gaming might be used through objective criteria. Some adolescents may feel that an amount of game playing, that other adolescents feel is not “excessive,” can be distressing for them. A subjective perception of “excessive” or “distressing” should be assessed in future studies. Finally, we did not consider different motivations, self-efficacy, and self-worth at the starting point between the pro-gamer students and IGD adolescents. These factors could lead a bias of the results in mental health and brain activity.

## 5. Conclusions

In response to long term internet game play, increased brain activity within the attention system was found in both groups. However, a well-organized support system, including a regular schedule and a supervisor, would lead to different results in terms of improved behavior scores and suppressing the brain activity within the orbitofrontal cortex.

## Figures and Tables

**Figure 1 ijerph-17-00441-f001:**
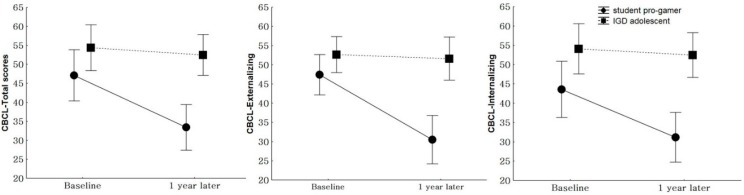
Comparison of the changes in the CBCL between student pro-gamer and IGD adolescent groups.

**Figure 2 ijerph-17-00441-f002:**
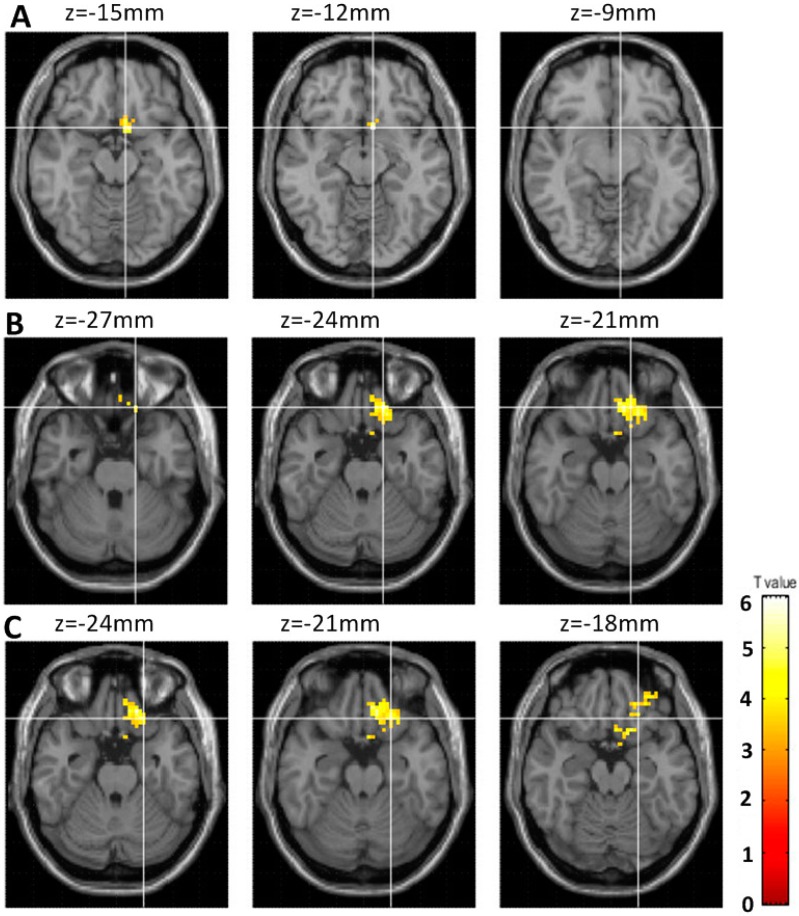
Regions showing differences in the changes of brain activity between the pro-gamer group and IGD adolescent group. (**A**) left subcallosal gyrus (x, y, z, −6, 12, −1), (**B**) left orbital gyrus (x, y, z, −15, 33, −24), (**C**) left inferior frontal gyrus (x, y, z, −21, 27, −21), yellow regions: the IGD adolescent group showed increased brain activity compared to the pro-gamers group.

**Table 1 ijerph-17-00441-t001:** Demographic and psychological data.

		Student Pro-Gamers	IGD Adolescents	Statistics
Age (years)		17.1 ± 0.3	16.5 ± 1.2	z = 1.07, *p* = 0.28
Education (years)		11.1 ± 0.3	9.7 ± 2.7	z = 1.63, *p* = 0.10
IQ		86.8 ± 6.3	90.4 ± 10.3	z = −1.29, *p* = 0.19
YIAS	B	56.3 ± 7.5	65.3 ± 7.5	z = −2.24, *p* = 0.02 *
	F	55.8 ± 13.0	61.4 ± 11.0	z = −1.46, *p* = 0.14
Game time (hours/day)	B	6.7 ± 1.4	7.0 ± 1.7	z = −0.24, *p* = 0.81
	F	7.2 ± 1.2	6.7 ± 2.0	z = 1.19, *p* = 0.23
CDI	B	7.8 ± 4.7	11.4 ± 5.6	z = −1.73, *p* = 0.08
	F	5.4 ± 3.9	12.2 ± 6.0	z = −3.10, *p* < 0.01 *
BAI	B	6.4 ± 2.9	7.9 ± 3.7	z = −1.07, *p* = 0.28
	F	6.2 ± 2.0	7.8 ± 3.3	z = −1.21, *p* = 0.81
K-ARS	B	12.5 ± 5.5	12.9 ± 5.8	z = −0.29, *p* = 0.22
	F	13.1 ± 3.8	12.7 ± 6.5	z = 0.22, *p* = 0.83
CBCL-T	B	47.1 ± 6.8	54.4 ± 13.9	z = −1.43, *p* = 0.15
	F	33.4 ± 8.9	52.5 ± 10.9	z = −3.53, *p* < 0.01 *
CBCL-E	B	47.4 ± 3.9	52.7 ± 11.3	z = −1.82, *p* = 0.07
	F	30.5 ± 7.8	51.6 ± 12.2	z = −3.81, *p* < 0.01 *
CBCL-I	B	43.6 ± 8.6	54.1 ± 14.4	z = −2.37, *p* = 0.02 *
	F	31.2 ± 7.6	52.4 ± 12.9	z = −3.93, *p* < 0.01 *

Notes: IGD adolescents: adolescents with internet gaming disorder (IGD), B: baseline, F: follow up; Young Internet Addiction Scale (YIAS), Child Behavior Checklist (CBCL), CBCL-T: total, CBCL-E: externalizing, CBCL-I: internalizing; Children’s Depressive Inventory (CDI), Beck Anxiety Inventory (BAI), Korean ADHD Rating Scale (K-ARS). * Statistically significant.

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
