# Peer review of "Comparison of Behavioral Changes and Brain Activity between Adolescents with Internet Gaming Disorder and Student Pro-Gamers"

_ijerph, 2020, doi:10.3390/ijerph17020441_

Round 1
Reviewer 1 Report
I read this manuscript with interest and I think the authors must be commended for addressing the relatively understudied population of pro-gamers. The manuscript can be considered for publication, should the authors be prepared to address some points:
in the passage at page 2, line 47, I suggest clarifying whether the cited literature is based on correlational or longitudinal studies (the latter could hypothesize causal links). at page 2, line 65 the authors state that "the initial onset of IGD was not associated with genre". Although I share this point, there is a large and accumulating literature on sex differences in this field and I think the authors should acknowledge at least a part of it. I suggest adding the reasons supporting the hypotheses, including appropriate references. In general, the introduction lacks completely an at least brief discussion of adolescence as a particularly at-risk developmental stage. I suggest adding this point to this section basing on previous literature (Cerniglia, Griffihts, Cimino et al., 2019; Cerniglia, Guicciardi, Sinatra et al., 2019; Cerniglia, Zoratto, Cimino, Laviola et al., 2017). In the Methods section: how were the subjects selected (line 116)? Most importantly: why did the subjects with IGD not receive any treatment? Why the other subjects did not accept to participate in the study (line 121)? Which were (briefly) the criteria for the acceptance in the High School pro-gamer department? This is interesting because one could wonder whether this is a very specific population. One of the most critical points is: why did the authors choose the CBCL as a measure to administer to adolescents? The CBCL is a tool used and validated for children (see the reference n. 30) not for adolescents. The Achenbach's measure usually used for adolescents is the Youth Self Report (a self-report measure; the CBCL is a report-form measure filled out by parents). A 6-18 version of the CBCL exsists, but it is filled out by parents. This point must be clarified. Lastly, in the discussion section I suggest discussing the fact that, although an "excessive" internet gaming is measurable through objective criteria, a subjective perception of what is "excessive" or "distressing" could have also been assessed. Some adolescents could feel that an amount of hours of gaming that for a fellow adolescent is not "excessive" isindeed distressing for him/her. This point could be addressed in the limitation section.Author Response
I read this manuscript with interest and I think the authors must be commended for addressing the relatively understudied population of pro-gamers. The manuscript can be considered for publication, should the authors be prepared to address some points:
1) in the passage at page 2, line 47, I suggest clarifying whether the cited literature is based on correlational or longitudinal studies (the latter could hypothesize causal links).
→ We revised the phrase as follows:
“Based on previous correlational studies, the brain neurobiology of IGD can be summarized as increased activity within the orbitofrontal cortex and decreased activity within the dorsolateral prefrontal cortex [9,10]. ”
2) at page 2, line 65 the authors state that "the initial onset of IGD was not associated with genre". Although I share this point, there is a large and accumulating literature on sex differences in this field and I think the authors should acknowledge at least a part of it.
→ We added the phrase of sex differences in IGD.
“Large, accumulating literature suggests that the prevalence of IGD is dominant in the male population [16]. ”
3) I suggest adding the reasons supporting the hypotheses, including appropriate references. In general, the introduction lacks completely an at least brief discussion of adolescence as a particularly at-risk developmental stage. I suggest adding this point to this section basing on previous literature (Cerniglia, Griffihts, Cimino et al., 2019; Cerniglia, Guicciardi, Sinatra et al., 2019; Cerniglia, Zoratto, Cimino, Laviola et al., 2017).
→ We added the phrase as follows:
“The period of adolescence was known as a particularly at-risk developmental stage for problematic internet game playing [15,16], because individuals were prone to impulsivity and uncontrolled, unplanned internet game play [16,17]. ”
4) In the Methods section: how were the subjects selected (line 116)? Most importantly: why did the subjects with IGD not receive any treatment? Why the other subjects did not accept to participate in the study (line 121)?
→ We revised the text describing how the subjects were selected as follows:
“From September 2016 to December 2017, 121 IGD adolescents visited the Department of Psychiatry at the OO University hospital for diagnosis and treatment. In contrast to the systematic caring group (student pro-gamer group), IGD patients who visited for an initial assessment but received no treatment (non-systematic caring group) were regarded as the compared group.
Of the 121 IGD adolescents, we found 27 adolescents who completed psychological and brain-imaging assessments at their first and second visits, but they had not received any treatment or interventions, such as cognitive behavior therapy or psychiatric medications, over the years. Although parents and caretakers had asked that the adolescents receive treatment for IGD, those adolescents refused treatment due to no interest in treatment, no insight of problematic behaviors, and laziness to come to the treatment center. On January 2018, we contacted 27 adolescents via phone to introduce our study, and 14 adolescents and their parents agreed to participate. The other 13 adolescents or their parents were not willing or able to participate in study.”
5) Which were (briefly) the criteria for the acceptance in the High School pro-gamer department? This is interesting because one could wonder whether this is a very specific population.
→ We revised the criteria about the requirements of acceptance to high school, as shown below:
“In 2017, 55 adolescents who wanted to be professional gamers applied to OO High School’s Pro-Gamer Department. With a ranking from the internet game ‘League of Legend’ competition and a basic academic ability test, 12 adolescents qualified and were accepted to be admitted to the school.”
6) One of the most critical points is: why did the authors choose the CBCL as a measure to administer to adolescents? The CBCL is a tool used and validated for children (see the reference n. 30) not for adolescents. The Achenbach's measure usually used for adolescents is the Youth Self Report (a self-report measure; the CBCL is a report-form measure filled out by parents). A 6-18 version of the CBCL exsists, but it is filled out by parents. This point must be clarified.
→ We used the “Manual for the Child Behavior Checklist/4-18 and 1991 profile.” We revised the phrase and references
“The CBCL is known as a screening tool for assessing problem behaviors in children and adolescents based on the parent’s self-report [33,34]. The Korean version of the CBCL (K-CBCL) has standardized reliability and validity [33,34]. Parents assessed their children and adolescents, aged between 4 and 18, using the K-CBCL, in terms of social adaptation and problem behavior. It consisted of 117 questions, with three subscales including a total problem score, as well as externalizing and internalizing scores. Higher scores indicated a greater degree of behavioral and emotional problems [33,34].”
4) Lastly, in the discussion section I suggest discussing the fact that, although an "excessive" internet gaming is measurable through objective criteria, a subjective perception of what is "excessive" or "distressing" could have also been assessed. Some adolescents could feel that an amount of hours of gaming that for a fellow adolescent is not "excessive" is indeed distressing for him/her. This point could be addressed in the limitation section.
→ We added it into the limitation section
“Third, the assessment tool for excessive internet gaming might be used through objective criteria. Some adolescents may feel that an amount of game playing, that other adolescents feel is not ‘excessive,’ can be distressing for them. A subjective perception of ‘excessive’ or ‘distressing’ should be assessed in future studies.”
Reviewer 2 Report
I like the ideas of the authors for trying to compare the brain activity and behaviors between student pro-gamers and IGD. I believe that this study will catch significant readers' attention and bring forth new ideas and attempts in future research. There are a few concerns that I have with the current manuscript.
(1)
Pg.1 Line 20 - 23. The authors stated that both groups displayed increased brain activity in the parietal lobe. Yet IGD adolescents showed increased higher brain activity within the left orbitofrontal cortex, including the subcallosal gyrus, orbital gyrus, and inferior frontal gyrus. Then, the authors concluded that gaming had a neutral effect on attention network's brain activity (line 24), what does that mean?
The authors had not discussed the positive and negative impacts of increased higher brain activity within the parietal lobe. Also, the authors should probably illust what are the harms by having increased higher brain activity within the left orbitofrontal cortex, subcallosal gyrus, orbital gyrus, and inferior frontal gyrus.
(2)
In Methods, line 110 -111, "However, pro-gaming students in the support system had the characteristics of a regular lifestyle while IGD adolescents without the support system showed an irregular lifestyle."
How did the authors know? Was that statement a general assumption? Was the statement as an observation of the participants that they had recruited in this study?
(3)
Line 116 - 117. What are the inclusion and exclusion criteria for participants? Please explained how the 27 were selected from 121 IGD patients?
(4)
Line 117 - 120. The 27 participants who had been selected to participate in the study had visited a hospital for problematic internet gameplay and completed psychological and brain imaging assessments at their first and second visits. However over the year, they did not receive any treatment or interventions. Was that part of the study? Was that intentional? What are the reasons that they did not receive any treatment or interventions? Was that ethical?
(5)
Line 121 - 124. The two groups of participants, 14/27 IGD adolescents/parents, and 12/12 pro-gamer students/parents. The samples are obviously very bias. IGD adolescents/parents and the pro-gamer students/parents could have very different motivations, self-efficacy and self-worth values at the starting points. The results could be biased, as it had a tendency to shift towards enlarging the differences between mental health and brain activity development after the study. What are the procedures that the authors had taken to minimize bias?
(6)
Table 1. (even with Figure 1)
Depression, anxiety and ADHD scores between the two groups were of no differences at the baseline nor the follow-up measurement. On the other hand, the baseline measurement for CBCL for both groups were obviously different. Parents rating on a child's problematic behaviors was significantly lower in the pro-gamer group than the IGD group. The further lowering score in follow-up, could it justifying bias because in the pro-gamer group, the children were prepared for a pro-gamer career, yet in the IGD group they have no clear future direction. I had not seen this element being discussed by the authors. What are the other possibilities for bias?
(7)
The authors need to explain the brain scan images better.
Figure 2.Comparison of fALFF changes between student pro-gamers and IGED adolescents over a year. A = subcallosal gyrus (z = -15mm, -12 mm, -9mm), B = left orbital gyrus, C = frontal gyrus. How do you show the brain scan images of 26 participants in Figure 2? Which pictures represent IGD? and which pictures represent pro-gamer?
(8)
Discussion. pg. 241-242. "eventhough both groups played internet games for more than 10 hours a day". Is that an assumption? because this was not shown in the results.
(9)
line 257 -258. Again, it can be parents' bias.
(10)
Line 273 - 284. What was the authors' original hypotheses? Was the results support the hypotheses? if not, discuss why. The current argument doesn't say anything about these/
(11)
Line 287 - 296. Doesn't connect with the results.
(12)
Line 297 - 309. Why dorsolateral prefrontal cortex is mentioned here? Was that related to the results? or the hypotheses?
(13)
Line 312 - 317. The bias is not only with the small numbers.
(14)
Conclusions were not sound.
Author Response
I like the ideas of the authors for trying to compare the brain activity and behaviors between student pro-gamers and IGD. I believe that this study will catch significant readers' attention and bring forth new ideas and attempts in future research. There are a few concerns that I have with the current manuscript.
(1) Pg.1 Line 20 - 23. The authors stated that both groups displayed increased brain activity in the parietal lobe. Yet IGD adolescents showed increased higher brain activity within the left orbitofrontal cortex, including the subcallosal gyrus, orbital gyrus, and inferior frontal gyrus. Then, the authors concluded that gaming had a neutral effect on attention network's brain activity (line 24), what does that mean?
The authors had not discussed the positive and negative impacts of increased higher brain activity within the parietal lobe. Also, the authors should probably illust what are the harms by having increased higher brain activity within the left orbitofrontal cortex, subcallosal gyrus, orbital gyrus, and inferior frontal gyrus.
→ We revised the abstract as follows:
“Both groups displayed increased brain activity in the parietal lobe (a component of the attention network) over the years. Compared to pro-gamers, IGD adolescents showed higher brain activity within the left orbitofrontal cortex. Brain activity within the orbitofrontal cortex was associated with CBCL-externalizing scores. These results suggest that gaming had increased the attention network’s brain activity, but a well-organized support system could lead to different results, in terms of improved behaviors and suppressing brain activity within the orbitofrontal cortex.”
(2) In Methods, line 110 -111, "However, pro-gaming students in the support system had the characteristics of a regular lifestyle while IGD adolescents without the support system showed an irregular lifestyle." How did the authors know? Was that statement a general assumption? Was the statement as an observation of the participants that they had recruited in this study?
→ We added it into the method section
“The pro-gamer students had school schedules, including regular academic classes (4 hours/day), physical sports class, strategy meetings, mealtimes, and game training time (3 hours/day in school). Two teachers managed and checked the schedules, while the parents of the IGD adolescents observed the life patterns of the adolescents and reported it.”
(3) Line 116 - 117. What are the inclusion and exclusion criteria for participants? Please explained how the 27 were selected from 121 IGD patients?
→ We revised our description of how the subjects were selected in the details.
“From September 2016 to December 2017, 121 IGD adolescents visited the Department of Psychiatry at the OO University hospital for diagnosis and treatment. In contrast to the systematic caring group (student pro-gamer group), IGD patients who visited for an initial assessment but received no treatment (non-systematic caring group) were regarded as the compared group.
Of the 121 IGD adolescents, we found 27 adolescents who completed psychological and brain-imaging assessments at their first and second visits, but they had not received any treatment or interventions, such as cognitive behavior therapy or psychiatric medications, over the years. Although parents and caretakers had asked that the adolescents receive treatment for IGD, those adolescents refused treatment due to no interest in treatment, no insight of problematic behaviors, and laziness to come to the treatment center. On January 2018, we contacted 27 adolescents via phone to introduce our study, and 14 adolescents and their parents agreed to participate. The other 13 adolescents or their parents were not willing or able to participate in study.”
(4) Line 117 - 120. The 27 participants who had been selected to participate in the study had visited a hospital for problematic internet gameplay and completed psychological and brain imaging assessments at their first and second visits. However over the year, they did not receive any treatment or interventions. Was that part of the study? Was that intentional? What are the reasons that they did not receive any treatment or interventions? Was that ethical?
→ As mentioned in answer (3), the data of the current study was obtained from patients recruited in a clinical setting. Although we recommended the treatment and intervention for the patients, we didn’t force the patients to complete treatment. We used the data of 27 patients from the patient pool.
(5) Line 121 - 124. The two groups of participants, 14/27 IGD adolescents/parents, and 12/12 pro-gamer students/parents. The samples are obviously very bias. IGD adolescents/parents and the pro-gamer students/parents could have very different motivations, self-efficacy and self-worth values at the starting points. The results could be biased, as it had a tendency to shift towards enlarging the differences between mental health and brain activity development after the study. What are the procedures that the authors had taken to minimize bias?
→ Frankly speaking, we didn’t consider that as a sample bias. We added it into the limitation section.
“Finally, we did not consider different motivations, self-efficacy, and self-worth at the starting point between the pro-gamer students and IGD adolescents. These factors could lead a bias of the results in mental health and brain activity.”
(6) Table 1. (even with Figure 1)
Depression, anxiety and ADHD scores between the two groups were of no differences at the baseline nor the follow-up measurement. On the other hand, the baseline measurement for CBCL for both groups were obviously different. Parents rating on a child's problematic behaviors was significantly lower in the pro-gamer group than the IGD group. The further lowering score in follow-up, could it justifying bias because in the pro-gamer group, the children were prepared for a pro-gamer career, yet in the IGD group they have no clear future direction. I had not seen this element being discussed by the authors. What are the other possibilities for bias?
→ We added to the discussion as follows:
“At baseline, CBCL-internalizing scores in the IGD adolescents were higher than those observed in student pro-gamer group. In addition, the student pro-gamer group showed improved behavioral scores as assessed using the CBCL, compared to IGD adolescent.…
With the CBCL results at baseline and follow up, a different interpretation can be suggested. The pro-gamer group was preparing for a pro-gamer career, while the IGD adolescent group had no clear future direction. This situation may have led to the bias of the parental assessment in the CBCL.”
(7) The authors need to explain the brain scan images better.
Figure 2.Comparison of fALFF changes between student pro-gamers and IGED adolescents over a year. A = subcallosal gyrus (z = -15mm, -12 mm, -9mm), B = left orbital gyrus, C = frontal gyrus. How do you show the brain scan images of 26 participants in Figure 2? Which pictures represent IGD? and which pictures represent pro-gamer?
→ We showed the difference in the changes of brain activity between the pro-gamer group and the IGD adolescent group (IGD adolescent – pro-gamers). We revised the title of Figure 2 and legend as follows:
“Figure 2. Regions showing differences in the changes of brain activity between the pro-gamer group and IGD adolescent group
Yellow regions: the IGD adolescent group showed increased brain activity compared to the pro-gamers group.”
(8) Discussion. pg. 241-242. "even though both groups played internet games for more than 10 hours a day". Is that an assumption? because this was not shown in the results.
→ We changed that phrase as follows:
“Even though both groups played internet games for more than 7 hours a day, it was assumed that …”
(9) line 257 -258. Again, it can be parents' bias.
→ As we responded in answer (6), we added the limitation of parent’s bias
(10) Line 273 - 284. What was the authors' original hypotheses? Was the results support the hypotheses? if not, discuss why. The current argument doesn't say anything about these.
→ Yes, those results support our hypotheses. To avoid confusion, we revised the hypotheses as follows:
“We hypothesized that long term internet game play would increase the brain activity within the attentional system in both groups. However, the existence of a support system, including a regular schedule and a supervisor, would lead to different results in terms of behaviors and brain activity. Student pro-gamers with a good support system would show improved behavioral scores compared to IGD patients. In addition, a good support system would prevent the hyperactivity within the orbitofrontal cortex in student pro-gamers, while a poor support system would not prevent the hyperactivity within the orbitofrontal cortex in response to impulsive internet game play.”
(11) Line 287 - 296. Doesn't connect with the results.
→ As mentioned in answer (10), that phrase was connected. For clearer understanding, we revised that phrase as follows:
“Increased brain activity, represented by the fALFF value, within the orbitofrontal cortex was only observed in IGD adolescents in this study. The imbalance between the dorsolateral prefrontal cortex and the orbitofrontal cortex, due to a hyper-activated orbitofrontal cortex, was thought to be related to cognitive control and decision-making dysfunctions, as well as impulsive behaviors in patients with gambling disorders [56]. In the seed base FC analysis, Kim et al. [57] reported the imbalance of FC from the orbitofrontal cortex to the dorsolateral prefrontal cortex in IGD patients. In internet addiction, functional dis-connectivity between the orbitofrontal cortex, dorsolateral prefrontal cortex, and anterior cingulate, due to hyper-activated orbitofrontal cortex, are associated with impulsive and delinquent behaviors [59].”
(12) Line 297 - 309. Why dorsolateral prefrontal cortex is mentioned here? Was that related to the results? or the hypotheses?
→ As discussed in answer (10) and (11), the dorsolateral prefrontal cortex was related to the hypothesis of the imbalance of the dorsolateral prefrontal cortex and orbitofrontal cortex.
(13) Line 312 - 317. The bias is not only with the small numbers.
→ Yes, we added more limitations
(14) Conclusions were not sound.
→ As the hypotheses were revised, we revised the conclusion as follows:
“In response to long term internet game play, increased brain activity within the attention system was found in both groups. However, a well-organized support system, including a regular schedule and a supervisor, would lead to different results in terms of improved behavior scores and suppressing the brain activity within the orbitofrontal cortex.”
Round 2
Reviewer 1 Report
The Authors were responsive to all comments and I think the paper can be published in the present form.